# Power Flow Management of Interconnected AC Microgrids Using Back-to-Back Converters

**Ezenwa Udoha** *[ID], **Saptarshi Das** [ID] and **Mohammad Abusara** *

Faculty of Environment, Science and Economy, University of Exeter, Penryn Campus, Penryn TR10 9FE, Cornwall, UK
* Correspondence: eu217@exeter.ac.uk (E.U.); m.abusara@exeter.ac.uk (M.A.)

**Abstract:** Microgrids have limited renewable energy source (RES) capacity, which can only supply a limited amount of load. Multiple microgrids can be interconnected to enhance power system availability, stability, reserve capacity, and control flexibility. This paper proposes a novel structure and control scheme for interconnecting multiple standalone microgrids to a common alternating current (AC) bus using back-to-back converters. The paper presents a high-level global droop controller that exchanges power between interconnected microgrids. Each microgrid considered in this paper comprises RES, battery, auxiliary unit, and load. The battery maintains the AC bus voltage and frequency and balances the difference in power generated by the RES and that consumed by the load. Each microgrid battery's charge/discharge is maintained within the safest operating limit to maximize the RES power utilization. To achieve balance and continuity of supply, renewable power curtailment and auxiliary power supplement mechanism is designed based on the bus frequency signalling technique. Performance evaluation shows that the proposed controller maximizes renewable power utilization and minimizes auxiliary power usage while providing better load support. The performance validation of the proposed structure and control strategy has been tested using MATLAB/Simulink.

**Keywords:** power flow management; interconnected microgrids; RES; back-to-back converter; droop control; global droop control

## 1. Introduction

The urgent need for action to combat climate change and its devastating impacts and secure a global net-zero emission by the middle of the century was echoed in the 27th Conference of the Parties (COP27) held recently in Egypt. The need to reach net-zero emissions by 2050 and reduce global $CO_2$ emissions by 45% from 2010 levels by 2030 has accelerated research advancements in renewable energy research. In the most recent International Energy Agency (IEA) report, it was found that about 775 million people globally have no electricity access in 2022, and most of this population is in sub-Saharan Africa [1,2]. Microgrids can provide clean, affordable, and sustainable energy in a centralised/decentralised network to meet energy-based electrification solutions. Therefore, RES, battery energy storage systems (BESS), and control system auxiliaries are coupled to form microgrids operating in standalone (autonomous) and grid-connected modes. Microgrids have limited RES capacity and can only supply a limited load, and increasing the load beyond the limit can lead to instability. Increased penetration of RES in low voltage microgrids with high R/X line impedance ratios can lead to real and reactive power coupling, voltage rise, and instability problems [3–7]. Contemporary literature, e.g., [7–11], has established various microgrid power management and controls. Still, the microgrid system stability, reliability, and efficiency decline as the load capacity and need for expansion increases.

Interconnected microgrids consist of two or more microgrids connected to maximise RES power utilisation and enhance power system availability, stability, and control flexibility. Interconnected microgrids can operate in standalone and grid-connected modes, just as

in a single microgrid. They possess better reserve capacity to improve network reliability, resilience, and sustainability [8]. Microgrids can be interconnected by either a common-coupling DC bus as in [9] or a common-coupling AC bus [10–12]. The common AC bus is a simple, easy-to-implement, proven, and cost-effective technology that allows for easy integration with existing AC links, transformers, power system auxiliaries and loads without further investment in power systems infrastructure, as would have been the case for a common DC link. Also, using power transformers helps form a more robust medium AC link, easing power transmission over long distances for power quality enhancement [13]. Due to excessive conversion equipment, interconnection with a common DC link is less reliable and available. Using a common DC link will require HVDC technology, a more complicated and sophisticated technology that still does not exist in the developing world. Using a common AC bus can avoid such complexities. Power electronic AC/DC/AC converters decouple two AC frequencies, and when properly controlled, the system can cope with undesirable disturbances that threaten system stability and robustness. It is simple to interconnect microgrids operating at the same voltage and frequency with static switches or breakers and a good synchronisation algorithm [5].

Many researchers have addressed microgrid interconnection in different structures and control topologies, focusing on microgrid interconnection with the common DC bus and tie-line representing the common AC bus. A robust distributed control for interconnected microgrids was designed in [14] to regulate the power flow among multiple microgrids in islanded mode. The microgrids are connected directly via a common bidirectional VSC-HVDC link which uses modal analysis and time-domain simulations to deal with critical issues that degrade the grid stability. Multiple microgrids are directly linked via a tie-line in [10–12]. In [15–18], microgrids are interconnected via a common DC link. A multi-microgrid power management system was proposed in [19], based on energy routers to handle network congestion and common issues in a multi-microgrid system. The system consists of a fixed grid connected to a voltage source converter (VSC), a circuit breaker, and an energy router with back-to-back converter technology connected in parallel with four microgrids. A distributed optimal tie-line power flow control for multiple interconnected AC microgrids in [20] consists of microgrids connected to the grid through a grid-tied switch. An optimal energy management strategy for minimising the operational cost of a multi-microgrid (MMG) network, proposed in [21], considered operation constraints and carbon emissions. An optimised framework for energy management of multi-microgrid systems, presented in [22], proposed a hierarchical energy management system for the optimal operation of multi-microgrids, which considered two-level optimisation. The research reported in [23] compares three different models of predictive control (MPC) coordination strategies based on decentralised, centralised, and hierarchically distributed MPC operations for interconnected home microgrids. A power management strategy for interconnected microgrids proposed in [11] uses power sharing frameworks and local objectives of multiple microgrids to maintain the power balance between generation and load. A power dispatch strategy for an interconnected, microgrids-based hybrid RES proposed in [24] ensures load demand in each microgrid is met through interaction with the utility grid. A new stochastic energy management technique for interconnected AC microgrids in [25] investigates the optimal operation and scheduling of interconnected microgrids with high penetration of RES—a framework based on the unscented transform (UT) method to model uncertainties. An energy management system for interconnected microgrids using the alternating direction method of multipliers (ADMM) strategy presented in [26] aims to interconnect microgrids to maximise the operational cost. A hierarchical decentralised system was proposed in [10] for the energy management of multiple microgrids. Despite the broadly published literature on interconnected AC microgrids, there is no clear report on interconnected microgrids with a common AC link. This specific area of research remains almost unaddressed, and no previous research has been able to provide this in sufficient detail.

This paper proposes a global droop control scheme that maximises RES power utilisation and provides better load support to manage power in interconnected microgrids. Traditional power transformers are provided to form a medium voltage AC (MVAC) bus, enabling maximum power transfer efficiency. A power electronic AC/DC/AC converter interconnects each microgrid to the common AC bus. The battery SoC of each microgrid is controlled to reflect deviation in AC bus frequency, indicating a surplus or shortage of power in the microgrid. Each microgrid prevents the charging/discharging of the battery SoC from exceeding its limits. It is essential to note that this study focuses on interconnected AC microgrids with no external grid connection. The main contribution is emphasised when microgrids are interconnected to a common AC bus. The simulation results are compared with that of independently operated microgrids. The controllers are implemented without a direct communication link between the interconnected microgrids. The main contributions of this paper can be highlighted as follows:

- This paper provides a novel structure and control topology of interconnected microgrid design for better RES utilisation and load support.
- Design of distributed controllers that limit the power demand of global converters by measuring each microgrid bus frequency deviation and adjusting its droop coefficient accordingly and in proportion to the bus frequency deviation. The controllers receive information about the bus frequencies using a low bandwidth communication link to enhance power flow among interconnected microgrids.
- Design of proposed global droop control mechanism for power management of multiple standalone interconnected AC microgrids. This controller ensures the right amount of power is exchanged between the interconnected microgrids.
- Design of RES power curtailment and auxiliary power supplement based on bus frequency signalling mechanism for power balance and continuity of supply.
- Performance evaluation of how well the suggested global droop controller satisfies the control priorities and design requirements following the limitations of the controllers. The evaluated results are compared based on three operating scenarios: (i) independent operation of multiple microgrids, (ii) multiple microgrids interconnected with the global droop control, and (iii) interconnected multiple microgrids operating with global droop control and global load.

The remaining parts of this paper are organised as follows. Section 2 illustrates the system structure, while the control strategy is discussed in Section 3. Simulation results are shown in Section 4, the performance evaluation is presented in Section 5, and the concluding section is presented in Section 6.

## 2. Overall System Structure of Interconnected Microgrids

The proposed structure of multiple standalone interconnected AC microgrids is represented in Figure 1. At the local microgrid level, when the microgrids are not interconnected, it functions as follows:

1. The BESS unit acts as a grid-forming unit controlling the local microgrid AC bus, as detailed in [27]. The BESS charging or discharging power depends on the difference between PV and load.
2. The PV-based RES is connected to the AC bus of the microgrid via a unidirectional DC/AC inverter, as detailed in [9]. The inverter controls output power depending on available irradiance.
3. The auxiliary unit consists of a unidirectional AC/DC converter that regulates the DC link voltage and the DC/AC inverter that controls power output based on the variation in AC bus frequency [27]. This auxiliary unit supports the BESS unit when the SoC is low and supplements power when the PV-based power cannot meet the load demand.

At the interconnected microgrid level, each microgrid is interfaced with the common AC bus via a back-to-back AC/DC/AC converter, and the system operates as follows:

1. The microgrid side AC/DC converter (local converter) regulates the DC link voltage.
2. The global bus DC/AC converter (global converter) regulates the power exchanged between the local microgrid and the rest of the system. However, all global DC/AC converters use conventional droop control to stay synchronised and to collectively control the global AC bus [9,10]. Power management at the global connecting DC/AC inverter employs a frequency signalling mechanism.

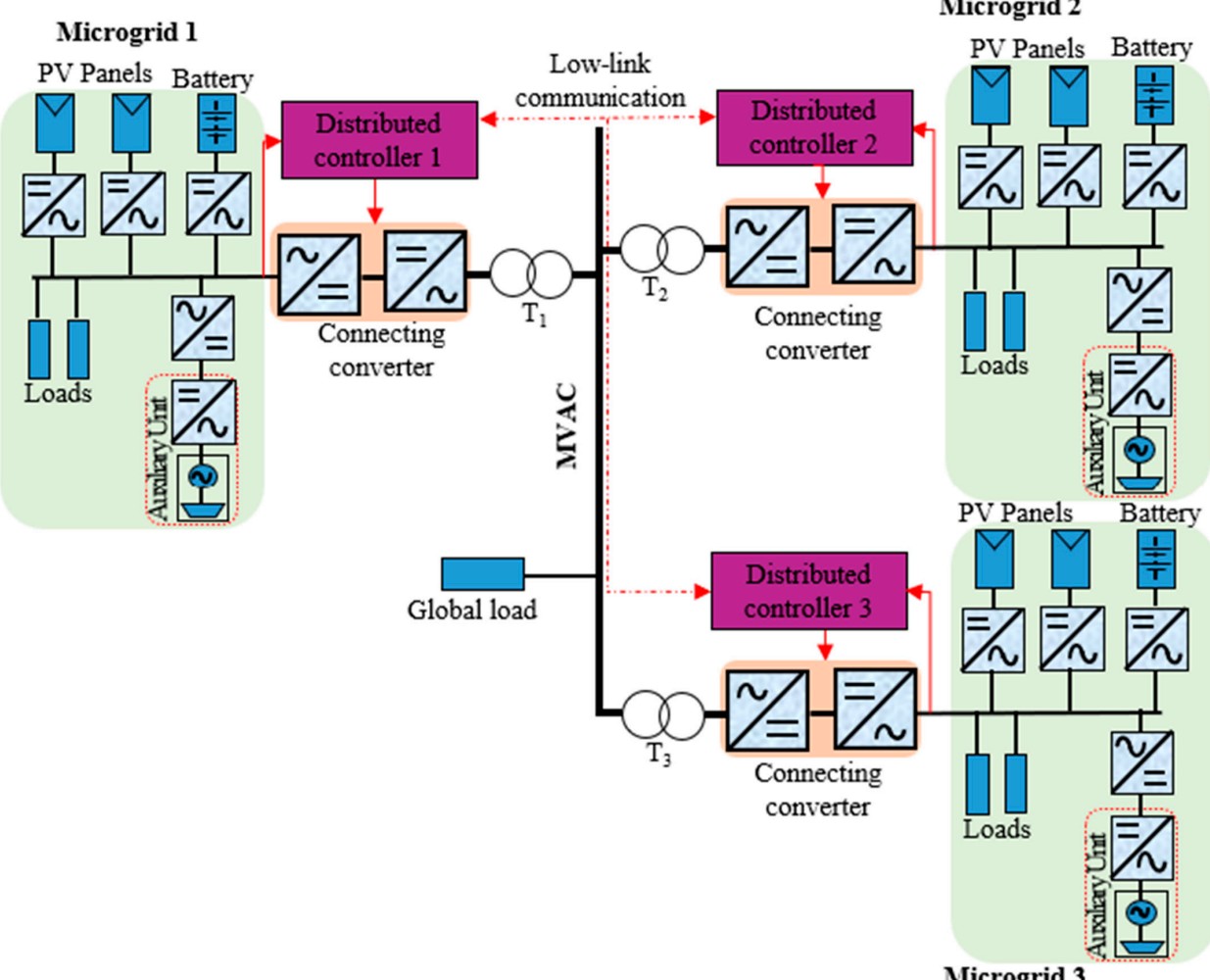

**Figure 1.** Proposed structure of multiple standalone interconnected microgrids.

## 3. Control Strategies of Different Components

Traditional droop control is used for all DC/AC converters. The conventional active Power-frequency ($P$-$\omega$) and reactive power-voltage ($Q$-$V$) droop control defined by Equation (1) are employed in all DC/AC converters. The primary controller in the active power droop strategy mimics the behaviour of the synchronous generator in terms of decreasing the frequency when the active power is increased and vice versa, as detailed in [27,28].

$$\begin{aligned} \omega &= \omega_0 - m(P - P^*) \\ V &= V_0 - n(Q - Q^*), \end{aligned} \tag{1}$$

where $\omega$, $V$ $\omega_0$, $V_0$, $m$, and $n$ represent the output frequency, the voltage amplitude, the nominal frequency, the nominal voltage, the frequency droop coefficient, and the voltage droop coefficient, respectively. Here, $P$ and $Q$ are the measured active and reactive power, while $P^*$ and $Q^*$ are active and reactive power setpoints, respectively.

### 3.1. Control of the BESS

The BESS DC/AC converter operates in voltage mode with a droop control coefficient set to zero. However, the frequency is varied as a signalling mechanism to control the other units in the system [24]. The output frequency of the BESS unit is given by $\omega = \omega_0 + \Delta\omega$, and the SoC determines the deviation in frequency $\Delta\omega$, according to Figure 2a. When the SoC is between $SoC_{low}$ and $SoC_{high}$, the frequency deviation $\Delta\omega = 0$. During this period, there is no need for export/import or curtailment/supplement. As the SoC increases, the frequency deviation increases until the SoC reaches $SoC_{max}$ where $\Delta\omega$ saturates. A similar trend exists for low SoC: when $\omega_0 < \omega < \omega_{high}$, the microgrid should export power to the rest of the system but with no PV curtailment. When $\omega_{high} < \omega < \omega_{max}$, the microgrid should curtail PV and export to the rest of the network. When $\omega_{low} < \omega < \omega_0$, the microgrid should import power from the rest of the network but with no supplement from the auxiliary unit. When $\omega_{min} < \omega < \omega_{low}$, the auxiliary unit should supplement the microgrid power and import power from the network.

### 3.2. Control of the Solar PV

When $\omega < \omega_{high}$, the PV unit should track the maximum power point depending on solar irradiance. Suppose that the microgrid frequency, as dictated by the BESS, deviated from its nominal value $\omega \neq \omega_0$ during steady state; the frequency of the PV inverter should also be $\omega$. According to [27], the *PV* power $P_{pv}$ is given by the expression in Equation (2):

$$P_{pv} = P_{pv}^* + \frac{\omega_0 - \omega}{m_p^{pv}} \tag{2}$$

Thus, the *PV* power will differ from the power setpoint $P_{pv}^*$ depending on the frequency deviation from its nominal value. The droop control can be modified to become a proportional-integral (PI) controller, as shown in Equation (3), to ensure that the power is the same as that of the setpoint, given as follows:

$$\omega = \omega_0 - \left( m_p^{pv} + \frac{m_i^{pv}}{s} \right) \left( P_{pv} - P_{pv}^* \right), \tag{3}$$

where $m_p^{pv}$ is the proportional droop control coefficient, $m_i^{pv}$ is the integral droop control coefficient, and $P_{pv}^*$ is the maximum power point ($P_{pv}^* = P_{MPPT}$). Depending on the microgrid frequency, the integral term will raise or lower the droop control curve. In Figure 2b, $\omega = \omega_0$ the PV power equals that of MPPT, i.e., $P_{pv} = P_{MPPT}$. In Figure 2c, the BESS frequency is increased, such as $\omega_0 < \omega < \omega_{high}$. This is the region where the microgrid should export power, not curtail *PV*. Therefore, the PI controller of the PV raises the droop curve (compared to that of Figure 2b), so the PV MPPT is maintained. In Figure 2d, the microgrid frequency is pushed higher, such as $\omega_{high} < \omega < \omega_{max}$. In this area, the microgrid should export power as well as curtail its PV output as described in Equation (4) $\omega > \omega_{high}$. The droop control becomes proportional only; the more the frequency increases, the less PV power is generated. When the frequency is $\omega = \omega_{max}$, PV power becomes zero. This can be described as the following:

$$
\begin{aligned}
\omega &= \omega_{max} - m_p^{pv} P_{pv}, & &\text{for } \omega > \omega_{high}, \\
\omega &= \omega_0 - \left( m_p^{pv} + \frac{m_i^{pv}}{s} \right) \left( P_{pv} - P_{MPPT} \right) & &\text{for } \omega < \omega_{high}.
\end{aligned} \tag{4}
$$

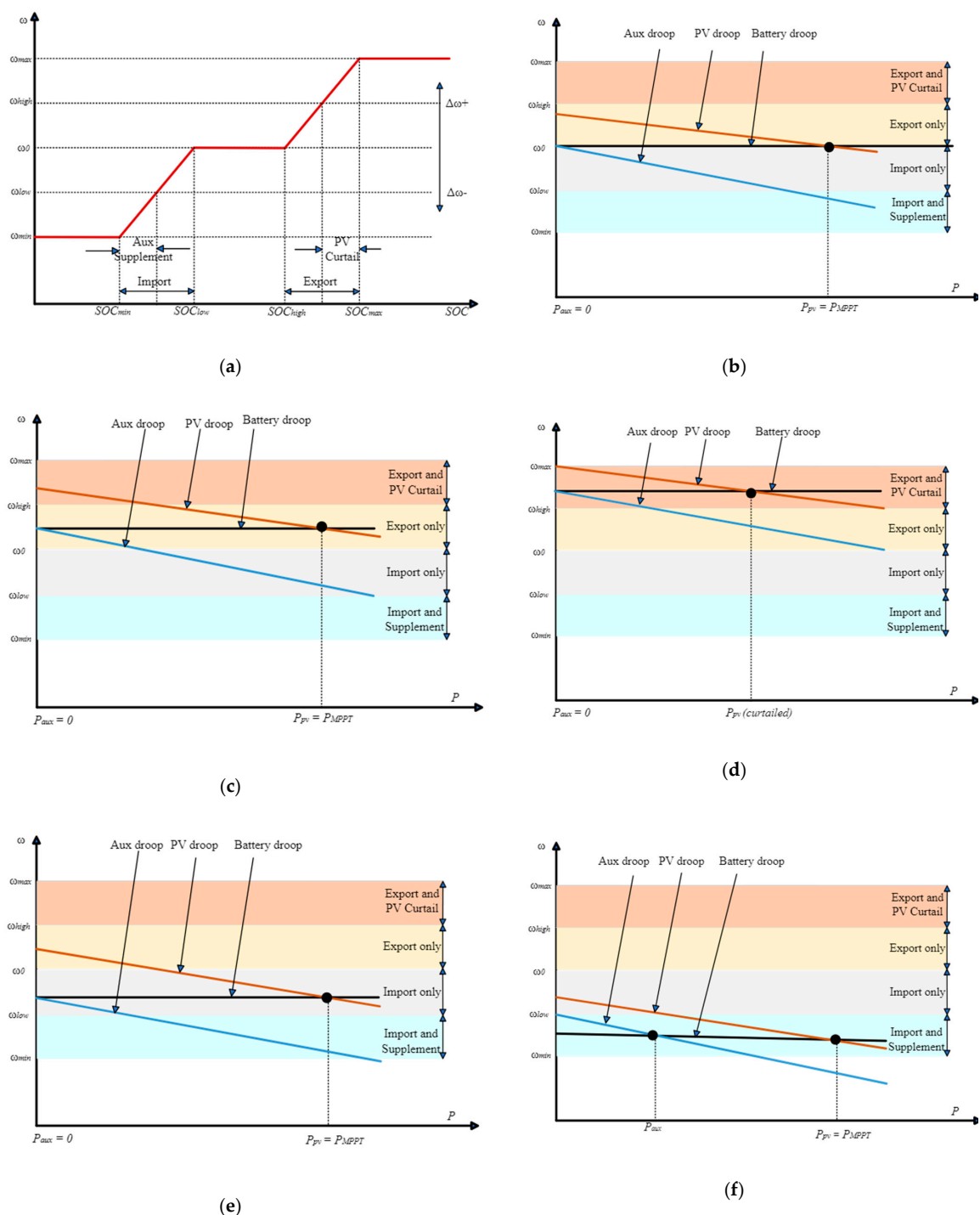

**Figure 2.** SoC, power–frequency deviation control curves. (**a**). SoC—frequency deviation control curve, power–frequency deviation droop control curves at (**b**). $\omega = \omega_0$, (**c**). $\omega_0 < \omega < \omega_{high}$, (**d**). $\omega_{high} < \omega < \omega_{max}$, (**e**). $\omega > \omega_{low}$, (**f**). $\omega < \omega_{low}$.

### 3.3. Control of the Auxiliary Unit

The purpose of the auxiliary unit is to supplement power whenever the SoC is low and import power from the rest of the network is insufficient. A fuel cell or micro gas turbine could power the auxiliary unit. There is no need for the unit to run unless the SoC and frequency start to go low. However, once the system is running, it should only produce power if the frequency is below $\omega_{low}$, as shown in Figure 2f. Otherwise, it should produce

zero power, as in Figure 2e. This can be achieved by using a PI controller if the frequency is higher than $\omega_{low}$, as shown in Equation (5):

$$
\begin{aligned}
\omega &= \omega_{low} - \left(m_p^{aux} + \frac{m_i^{aux}}{s}\right)P_{aux} \quad \text{for } \omega > \omega_{low}, \\
\omega &= \omega_{low} - m_p^{aux}P_{aux} \qquad\quad \text{for } \omega < \omega_{low}.
\end{aligned}
\tag{5}
$$

### 3.4. Control of the Interconnecting Back-to-Back Converter

The microgrid-side converter (local converter) regulates the DC link voltage by setting the power setpoint of the droop control, as seen in Equation (7):

$$
\omega = \omega_0 - m_p^{local}(P_{local} - P_{local}^*),
\tag{6}
$$

$$
P_{local}^* = \left(k_p^{vdc} + \frac{k_i^{vdc}}{s}\right)(V_{dc} - V_{dc}^*),
\tag{7}
$$

where $k_p^{vdc}$ is the proportional droop control coefficient, $k_i^{vdc}$ is the integral droop control coefficient, $V_{dc}^*$ is the DC link voltage setpoint, and $V_{dc}$ is the measured voltage of the DC link. However, there is no need for an integral controller in the droop control because the power setpoint $P_{local}^*$ is an output of a PI controller. Hence, it can compensate for any frequency deviation, i.e., the integral term in Equation (7) will raise the droop curve up or down depending on the microgrid frequency. The MVAC side converter (global converter) is responsible for the power exchange between the microgrid and the rest of the network. All the global converters control the MVAC collectively via droop control on the global sides, as described in Equation (8):

$$
\omega_{global} = \omega_0 - m_p^{global}\left(P_{global} - P_{global}^*\right)
\tag{8}
$$

The global converter needs to export or import power depending on the status of the microgrid's local frequency and hence the SoC of its battery. Therefore, the power setpoint of the *global* converter $P_{global}^*$ is given by Equation (9):

$$
P_{global}^* = k \times \Delta\omega_{local}
\tag{9}
$$

The power-sharing between the global converters depends on the power setpoints and loads on the MVAC bus, as shown in the next section.

### 3.5. Steady State Power Flow between the Interconnected Microgrids

As explained above, all the global inverters are set to operate in droop control with their power setpoints set according to their microgrids battery SoC. To have a balanced system with no control signals between the microgrids, no integral term should be used in the droop control. If an integral term is used, the power output from the global inverter (power export/import from the microgrid) must equal the power setpoint. To have a balanced system, these setpoints must be determined collectively by a centralised controller. However, if a proportional controller is used, then an increase in the power setpoint will increase the power export and the exact amount of power exchange will depend on the power setpoints of the other global inverters and the load connected to the global bus. The total power dissipated by the load should equal the output power generated by the multiple global converters, as described in Equation (10):

$$
P_L = \sum_{i=1}^{N} P_{\exp,i}
\tag{10}
$$

At the steady state, all global converters must operate at the same bus frequency. Assuming that all global converters have the same droop coefficient, the global frequency is given in Equation (11):

$$\omega_{global} = \omega_0 - \frac{m_p^{global}}{N} \sum_{i=1}^{N} \left( P_{exp,i} - P_{exp,i}^* \right) \tag{11}$$

where $N$ is the total number of connecting global converters.

Substituting Equation (10) into Equation (11) gives Equation (12):

$$\omega_{global} = \omega_0 - \frac{m_p^{global}}{N} \left( P_L - \sum_{i=1}^{N} P_{exp,i}^* \right) \tag{12}$$

Substituting Equation (12) into the active power–frequency component of Equation (1) gives Equation (13):

$$P_{exp,i} = \frac{P_L}{N} + P_{exp,i}^* - P_{exp,avg}^*, \tag{13}$$

where

$$P_{exp,avg}^* = \frac{\sum\limits_{i=1}^{N} P_{exp,i}^*}{N} \tag{14}$$

The expression in Equation (13) shows that if all units have the same power setpoint, all units will supply the global load equally and there will be no power exchange between the microgrids. It also shows that each unit will supply its portion of the global load plus a term equal to the difference between that unit's power setpoint $P_{exp,i}^*$ and the average of all setpoints $\sum P_{exp,i}^*$. Additionally, if the average of all setpoints is greater than $P_L/N + P_{exp,i}^*$, the microgrid will import rather than export power. The above equations are represented in Figure 3 which shows the system model and its high level controller.

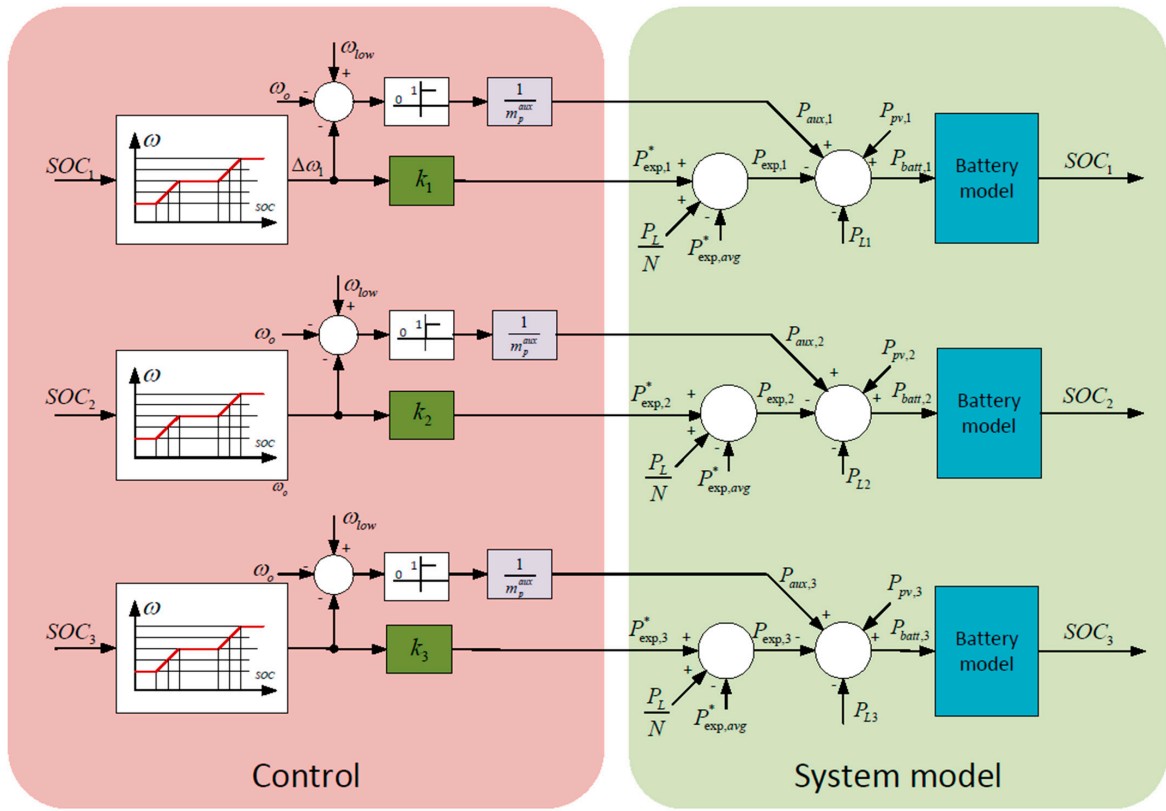

**Figure 3.** High-level control of multiple standalone interconnected microgrids.

## 4. Simulations and Results

### 4.1. Details of the Simulation Set-Up

A detailed model for three interconnected microgrids was built in Matlab/Simulink. Each microgrid consists of PV-based RES unit, BESS, Auxiliary unit and load. The system parameters used in the simulation are listed in Table 1. The simulation results are presented in Figure 4.

**Table 1.** System parameters used in the simulation.

| Parameter | Symbol | Value |
|---|---|---|
| Nominal bus frequency | $\omega_0 = 2\pi f_0$ | 314 rad/s |
| Nominal bus voltage | $V_0$ | 230 V |
| Nominal DC link voltage | $V_{dc}$ | 750 V |
| DC link voltage P-controller gain | $k_{p\_dc}$ | 20 |
| DC voltage I—controller gain | $k_{i\_dc}$ | 60 |
| DC link capacitor | $C_{dc}$ | 1200 $\mu$F |
| Active power droop coefficient | $m$ | $0.9 \times 10^{-4}$ rad/s/W |
| Reactive power droop coefficient | $n$ | $0.9 \times 10^{-4}$ V/Var |
| Frequency gain | $k$ | 30,000 |

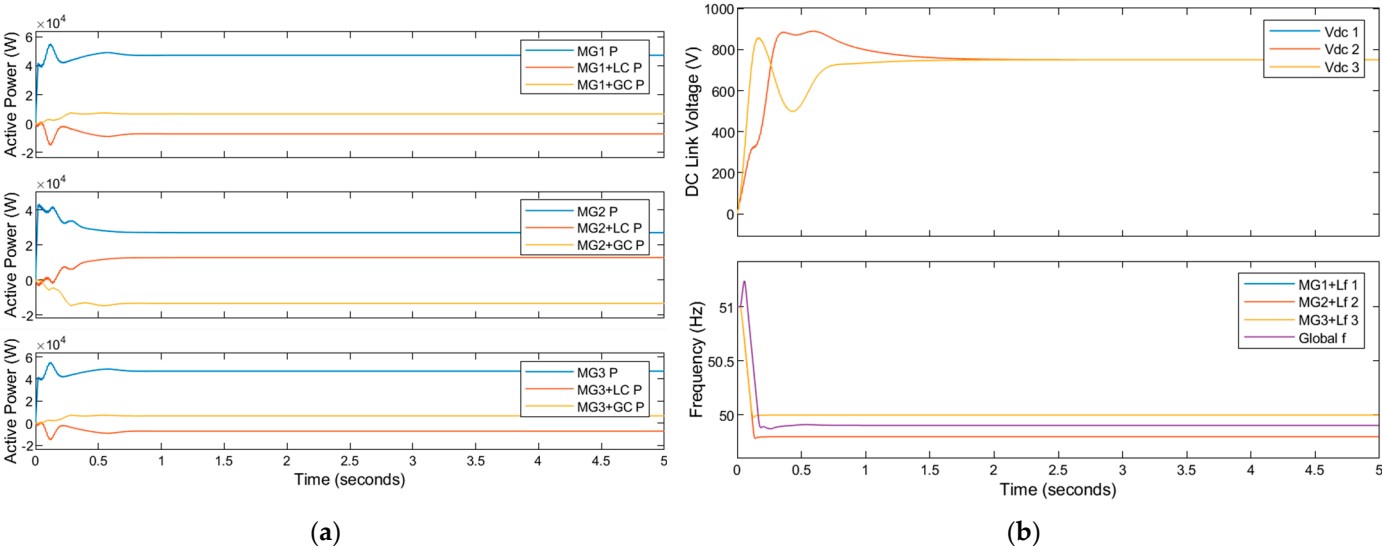

(**a**)  (**b**)

**Figure 4.** Response of the interconnected microgrids operating at varied frequencies and load: (**a**) power utilisation, (**b**) operating DC link voltage and frequency.

Figure 4a shows simulation results of interconnected microgrids operation at 50 Hz, 49.80 Hz, and 50 Hz frequencies and loads of 40 kW, 25.8 kW, and 40 kW for microgrids one, two, and three, respectively. The variation in frequency in microgrid two reflects a power shortage of about 14.2 kW. The power shortage in microgrid two is equitably supplemented with 7.1 kW of power each from microgrids one and three, respectively, following a frequency deviation below the nominal of microgrid two. The global frequency is maintained at 49.9 Hz against the different microgrid operations at a frequency deviation below the nominal of microgrid two. The varying nature of the different microgrid and global bus frequency values is shown in Figure 4b. Hence, the DC link voltage is controlled at 750 V, as seen in Figure 4b. However, the DC link voltage of the local converters for microgrid two reaches a steady state at about 1.6 s, while the global bus frequency is maintained at 49.90 Hz from about 0.6 s.

### 4.2. High-Level Simulation of Interconnected Microgrids

Due to the slow simulation of the detailed model and the need for testing the system over a longer period, the high-level system represented in Figure 3 was created in Matlab/Simulink which depicts the multiple interconnected microgrids consisting of PV-based RES unit, BESS, Auxiliary unit and load with the proposed droop controller. Each microgrid power is balanced, as shown in Equation (15):

$$P_{batt,i} = P_{pv,i} + P_{aux,i} - P_{L,i} - P_{exp,i}, \tag{15}$$

where $i$ represents the number of microgrids, $P_{batt}$ is the battery power, $P_{pv}$ is the PV-based RES power, $P_{aux}$ is the auxiliary power, $P_L$ is the load, and $P_{exch}$ is the power exchanged (import/export). The system parameters are described in Table 2.

**Table 2.** System parameters for high-level simulation.

| Parameter | Symbol | Value |
|---|---|---|
| Nominal bus frequency | $\omega_0 = 2\pi f_0$ | 314 rad/s |
| Battery capacity | $C_1 = C_2 = C_3$ | 2000 mAH |
| Maximum SoC | $SoC_{max}$ | 100% |
| Minimum SoC | $SoC_{min}$ | 30% |
| Low SoC | $SoC_{low}$ | 40% |
| High SoC | $SoC_{high}$ | 90% |
| Global drooping coefficient | m | $1 \times 10^{-4}$ rad/s/W |
| Maximum frequency deviation | $\Delta\omega_{max}$ | 1 |
| High-frequency deviation | $\Delta\omega_{high}$ | 0.1 |
| Reference power | $P_{ref\_max}$ | 1000 W |

From the simplified high-level model, each microgrid operates to meet the demand of its primary load. However, the results of the high-level simulation model of multiple interconnected microgrids are shown in Figures 5–7.

The first high-level simulation results represent the output response of three microgrids operated independently within a minimum and maximum SoC of 30% and 100%, respectively. Figure 5a shows the power output of the PV-based RES and the curtailed RES, the auxiliary unit, the SoC, and the load power of microgrid one. The general expectation from this simulation scenario is that any excess power should be curtailed, and priority should be given to the full utilisation of the PV-based RES to feed the load. In microgrid one, the RES power supply is greater than the load demand. The RES is used to supply power to the load, and the surplus from the RES is curtailed. The auxiliary unit does not supply any power and remains on standby as the battery SoC goes up to its full limit for the day. At about time $t = 4$ h, the SoC tends to decrease due to a slight increase in the load. At $t = 19$ h, the SoC reflects the gradual increase in load demand with a gradual slight decline. However, the battery power is provided at all times to balance the system as priority is given to full utilisation of PV-based RES, the SoC is high, and the auxiliary unit only supplies power when needed. Figure 5b shows the simulation result for microgrid two, which illustrates that the power generated from the PV-based RES exceeds the load demand. The auxiliary unit is on standby and does not supply any power as the battery SoC goes up to its full limit for the day. At about $t = 4$ h, the SoC slightly decreases due to a sharp increase in the load. At $t = 19$ h, the SoC reflects the gradual increase in load demand with a slight decline, which goes back up at $t = 21$ h. Figure 5c shows the simulation results for the third microgrid, indicating that the load demand is greater than the available RES power. The PV-based RES supplies its full capacity to the load, which is insufficient to meet the demand of the load, and the auxiliary unit supplies the difference to meet the load demand. The auxiliary unit starts to supply power at $t = 0$ h, as the battery SoC goes down to its low limit of about 32%. The available power from the PV-based RES is less than the load, and this caused the SoC to stay within its low limit, which triggers supply from the

auxiliary unit. At about time $t = 12$ h, the SoC goes up and then down at $t = 15$ h, due to intersecting with the load demand and available RES curves, and at this point, the auxiliary unit supplies zero power. The auxiliary unit supplies power to meet the load demand for the rest of the simulation time. Figure 5d shows the 30 days simulation for SoC operated under different load and RES profiles, and the results illustrate that the SoC remains within its limits. The controller prevents the SoCs from exceeding their maximum and minimum limits.

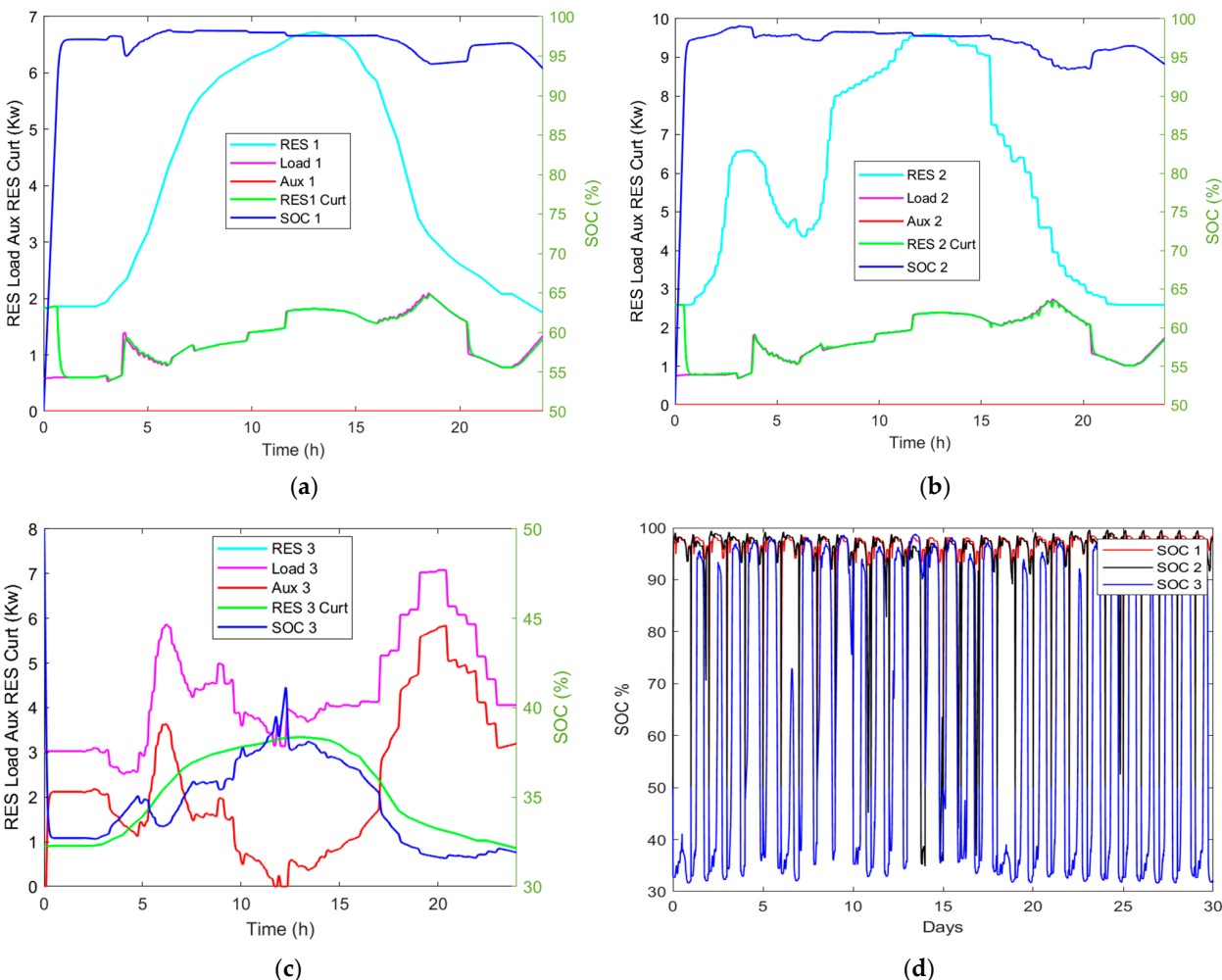

**Figure 5.** (**a**–**d**). Output responses for multiple microgrids independently operated for 30–100% SoC with no global droop controller: (**a**) microgrid one, (**b**) microgrid two, (**c**) microgrid three, and (**d**) SoC.

The second simulation scenario represents the results of three microgrids interconnected with the proposed global droop controller, operating within a minimum and maximum SOC of 30% and 100%, respectively. Figure 6a shows the output power of the PV-based RES and the curtailed RES, the auxiliary unit, the SoC, the load, and the power exported for microgrid one. After power export, it is anticipated that any excess RES power should be curtailed, and the available PV power should be used to supply the load. In the case of insufficient supply from the RES, power should be imported from other microgrids with surplus power. The auxiliary unit supplies the shortage if the power export/import is insufficient to meet the load demand. The available RES power in microgrid one exceeds the load demand throughout the simulation. The RES profile supplies the power for the load, some RES power is fairly exported via $P_1$, and the surplus power is curtailed. The battery SoC goes up to its full limit most of the time, and the auxiliary unit is on standby and does not supply any power. At about $t = 19$ h, the SoC goes below 95%, which cor-

responds to the highest point of the load, and the SoC goes back up to 95% as the load reduces. Figure 6b shows that the available RES power in microgrid two is greater than the load, indicating power export from microgrid two to deficient microgrids via the $P_2$. The RES supplies power to the load, the surplus power is fairly exported via $P_2$, and the remaining excess is curtailed. The battery SoC goes up and stays within its full limit, keeping the auxiliary unit on standby and not supplying. At about $t = 18$ h, the SoC slightly reduces below 95%, and this slight reduction of SoC corresponds to the highest point of load demand, and the SoC goes back up to 95% as the load demand and the curtailed RES power reduce. Figure 6c shows that the available PV power is less than the load demand in microgrid three, which implies the need for power import via $P_1$. The PV-based RES power is fully utilised to supply the load, which is insufficient to meet the demand. The two other microgrids equitably supply the remaining power shortage starting from $t = 0$ h. Due to the lower available PV, the SoC stays within its lowest limits at intervals between $t = (0–10)$ h and (16–24) h, which causes the auxiliary unit to supply power during those intervals to meet demand. The auxiliary unit supplies power as required and stays on standby within $t = (11–16)$ h, as the SoC goes above up above the low limit. The SoC goes up to its high limit due to the load demand being met by the available RES and power equitably imported from the first and second microgrids. Figure 6d shows the 30 days simulation results for the SoC operated under different loads and RES conditions. The results show that despite the intermittent nature of the RES and changes in load requirements, the SoC maintains its boundaries. The maximum charging and minimum discharging levels of the battery are preserved. Figure 6d shows the frequency curve at the global bus with the global droop controller. The curve shows that frequency is maintained within its operating limits at the global bus, irrespective of the intermittent nature of the RES power and variations in the load demands.

The third simulation case represents the output responses of three microgrids interconnected with a global droop controller and global load, operating within a minimum and maximum SoC of 30% to 100%, respectively. Figure 7a shows the available PV-based RES and the curtailed RES power, the auxiliary unit, the SoC, the load profile, and the power exported for microgrid one. The available RES power is greater than the load, and the RES supplies power to meet the load and export power via $P_1$, and the surplus from the RES is curtailed. The auxiliary unit is on standby and does not supply power as the battery SoC significantly goes up to its full limit. At about $t = 19$ h, the SoC slightly goes down to about 85% while the system exhausts its available RES power due to increased load demand. At about $t = 21$ h, the battery SoC instantly goes back up to its full limit due to a sharp reduction in load. Figure 7b shows that the available RES power is greater than the load demand; therefore, power is exported via $P_2$, and the surplus is curtailed. The auxiliary unit is on standby and does not supply power while the battery SoC goes up to its full limit. At about $t = 18$ h, the battery SoC gradually decline to about 85%, indicating a maximum utilisation from the available RES at the highest point of the load demand. Figure 7c shows that the available PV power is less than the load demand, indicating the need for power import into the microgrid via $P_3$. Due to an insufficient supply of RES power, the SoC goes down to its low limit, and the auxiliary unit starts to supply power at $t = 0$ h, while the available PV-based RES power is fully utilised to supply the load, and no RES is curtailed. At $t = 12$ h, the SoC momentarily goes up and down due to the point of intersection of the RES and load demand curve. After that, the SoC further goes down to its low limit, and the auxiliary unit supplies more power. Figure 7d shows that the global load demand at the global bus is equitably fulfilled at every instant such that the combined effect of the supply of the global load follows the demand. Figure 7e shows the 30 days simulation results for the SoC operated with the global droop controller and global load under different microgrid loads and RES profiles. The result illustrates that the SoC remains within its limits regardless of the variation in the supply of RES and changing local and global load demands. However, the maximum charging and minimum discharging levels of the battery are preserved throughout the simulations. Figure 7f shows the global frequency curve of the

global bus operated with the global droop controller and global load. The curve illustrates that the frequency of the global bus is maintained within its operational limits, irrespective of the variations in RES power and local and global load demands.

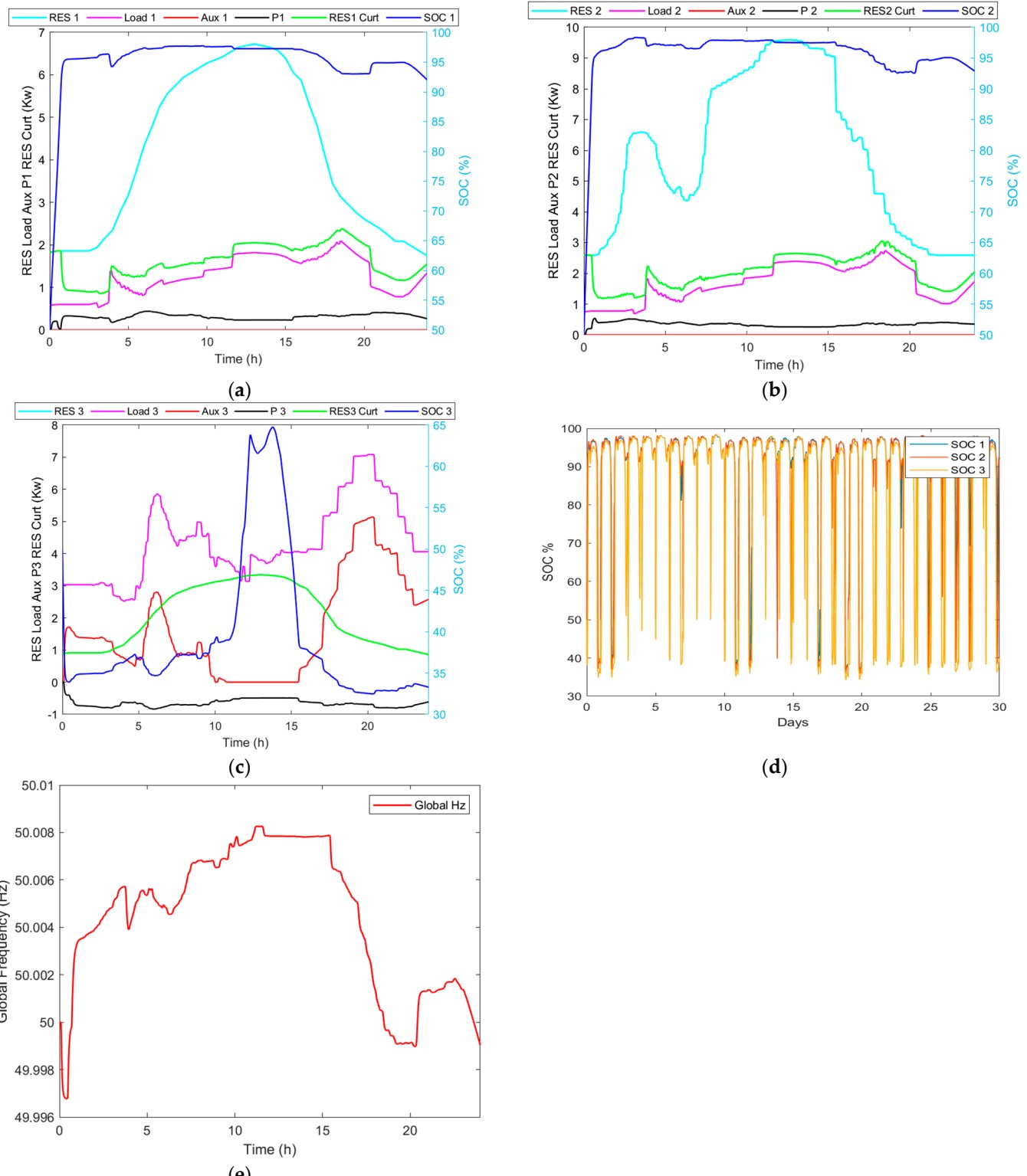

**Figure 6.** (**a**–**e**). Output responses for multiple microgrids interconnected with global droop controller for 30–100% SoC: (**a**) microgrid one, (**b**) microgrid two, (**c**) microgrid three, (**d**) SoC, and (**e**) global frequency.

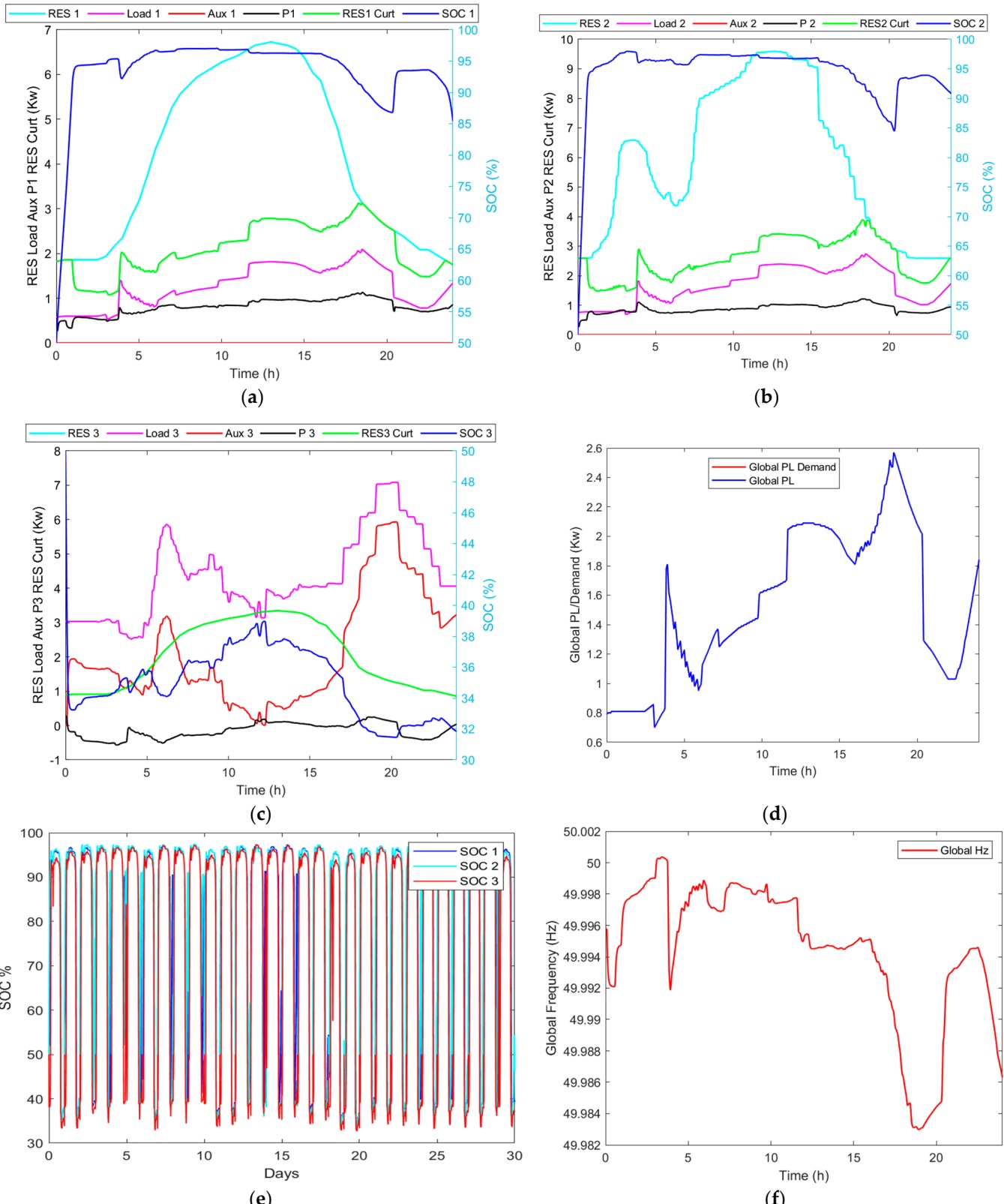

**Figure 7.** (**a**–**f**). Output responses for multiple microgrids interconnected with global droop controller and load for 30–100% SoC: (**a**) microgrid one, (**b**) microgrid two, (**c**) microgrid three, (**d**) global load, (**e**) SoC, (**f**) global frequency.

## 5. Performance Evaluation

It is important to reiterate that the main focus of this paper on power flow management of interconnected AC microgrids using back-to-back converters is to maximise RES power utilisation and provide incredible support for the load. This study examines the outcomes of independently operating AC microgrids, both standalone and when they are interconnected using our proposed technique, based on how much RES power is utilised and how much auxiliary power is supplemented to meet the demand of the load. Hence, this section compares simulation results for 30 days based on three operating scenarios:

1. Independent operation of the microgrids;
2. Interconnected operation of multiple microgrids with the proposed global droop control;
3. Interconnected operation of multiple microgrids with the proposed global droop control and global load. The performance of the three operating scenarios is assessed using a series of simulation data, and the results are shown in the subsequent sessions.

Figure 8 compares the RES power curtailment simulation results evaluated over 30 days based on the three operating scenarios. The results show that in all 30 days, the most available RES power is curtailed each day when the microgrids are independently operated compared with when multiple microgrids are interconnected with the proposed global droop controller with or without the global load in place.

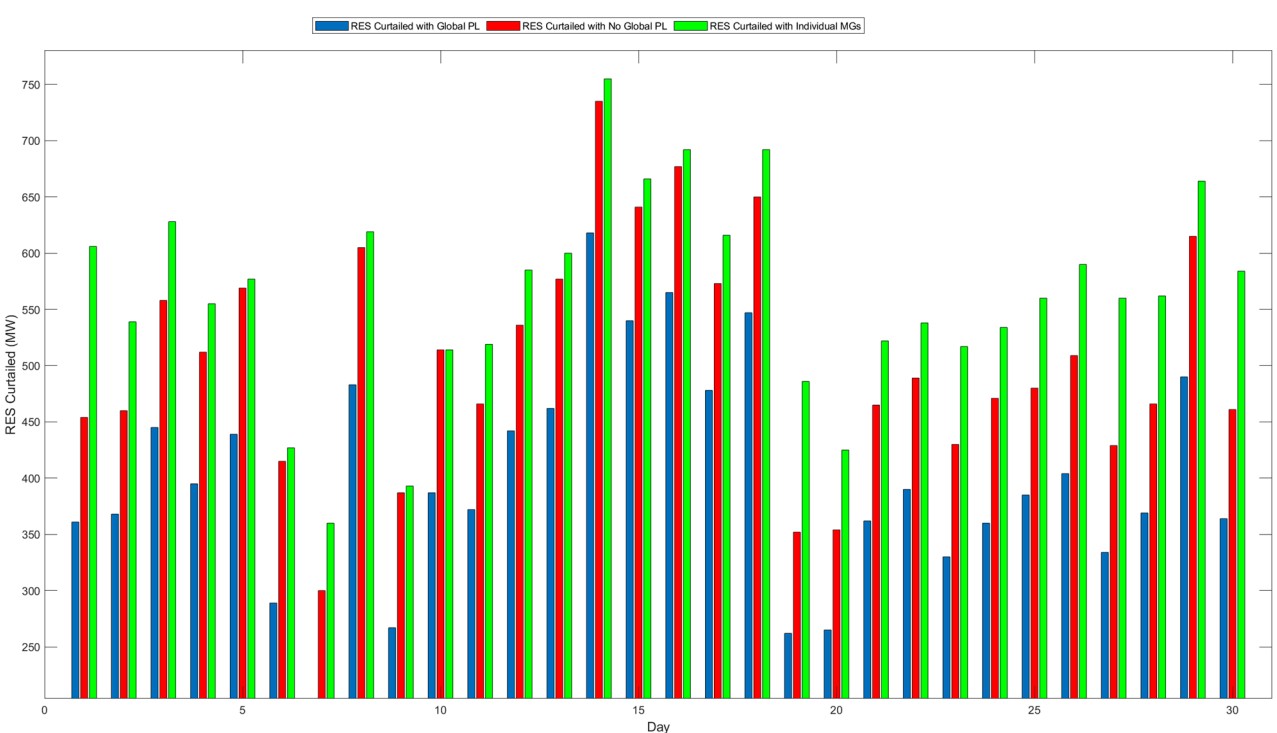

**Figure 8.** Comparing RES power curtailed with individually operated microgrids, with the proposed global controller and with the proposed controller and global load.

Figure 9 compares the auxiliary power utilisation for 30 days of simulation results and is evaluated based on the three operating scenarios. The results show that the highest amount of auxiliary power is supplemented daily when the microgrids are independently operated compared to when multiple microgrids are interconnected with the proposed global droop controller with or without the global load. The smallest amount of auxiliary power utilisation occurs when multiple microgrids are interconnected with the proposed global droop controller. However, there is a gradual increase in auxiliary power utilisation when the global load is connected to the global bus.

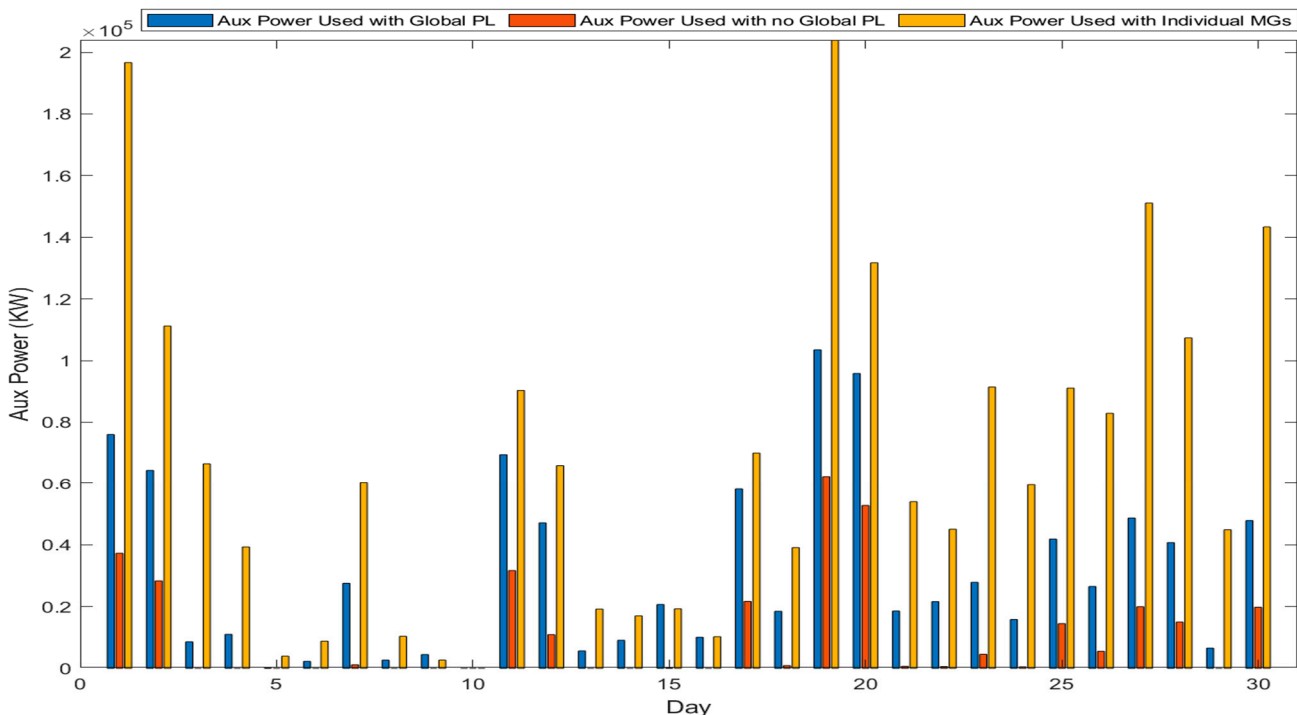

**Figure 9.** Comparing auxiliary power utilised for individually operated microgrids, with the proposed global controller and with the proposed controller and global load.

Figure 10a compares the total RES energy curtailed from overall simulation results evaluated based on the three operating scenarios. The results show that the maximum available RES energy of about 405 MWh is curtailed when the microgrids are independently operated compared to about 364 MWh of RES energy curtailed when multiple microgrids are interconnected with no global load. However, the least RES energy of about 287 MWh is curtailed when the global load is connected to the proposed control strategy. This implies that more RES energy is utilised with the proposed technique as the global load is connected to the global bus. Figure 10b compares the total energy utilised from the auxiliary unit evaluated based on the three operating scenarios. The results show that the highest energy of about 48,883 MWh is utilised from the auxiliary unit when the microgrids are independently operated compared to the smallest of about 7876 MWh obtained when multiple microgrids are interconnected with the proposed global droop controller. More energy, about 22,344 MWh, is utilised from the auxiliary unit when multiple microgrids are interconnected with the proposed global droop controller with the global load. Hence, there is a gradual increase in energy utilisation from the auxiliary unit when the global load is connected to the global bus.

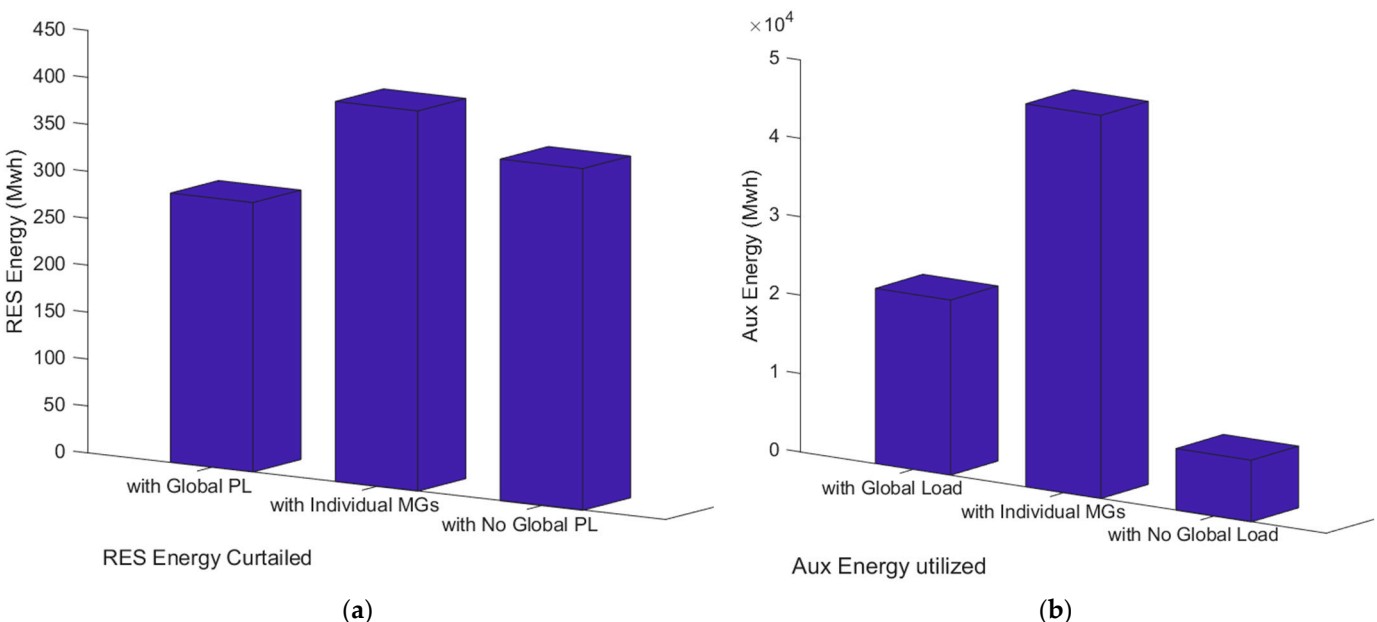

**Figure 10.** Comparing total energy curtailed and auxiliary energy utilised for individually operated microgrids and with the proposed technique. (**a**) RES energy curtailed, (**b**) Auxiliary Energy utilized.

## 6. Conclusions

The proposed novel structure and power flow management system for multiple standalone interconnected AC microgrids using back-to-back converters and traditional power transformers to form a common AC bus have been formulated and tested in Matlab/Simulink. The proposed controller combines the global droop controller and the frequency bus-signalling techniques to manage the power flow between interconnected microgrids. The controller is implemented without any communication link between the microgrids based on the local and global droop controllers and varying the AC bus frequency within allowable standards. The auxiliary unit is left to stay on standby and only supply power when needed, depending on the deviation in frequency and to avoid frequency degradation below allowable limits. In contrast, the RES power curtailment mechanism is utilised to curtail the RES when required to achieve balance and continuity of supply. However, our results demonstrate the effectiveness and efficiency of the proposed interconnected microgrid structure and global droop controller, which is capable of maintaining the system requirements within the defined limitations. We next focused on different optimisation approaches to determine the optimal global droop control strategy for the structure of multiple interconnected standalone AC microgrids.

**Author Contributions:** Conceptualization, E.U., S.D. and M.A.; methodology, S.D. and M.A.; software, E.U. and M.A.; validation, S.D. and M.A.; formal analysis, E.U. and M.A.; investigation, E.U.; resources, M.A.; data curation, E.U.; writing—original draft preparation, E.U.; writing—review and editing, S.D. and M.A.; visualization, E.U.; supervision, S.D. and M.A.; project administration, M.A.; funding acquisition, E.U. All authors have read and agreed to the published version of the manuscript.

**Funding:** This research was funded by Tertiary Education Trust Fund (TETFUND) under the Federal University of Petroleum Resources, Effurun, Nigeria AST&D 2018 Intervention, grant number FUPRE/TO/AST&D/2018.

**Data Availability Statement:** The data is available from the corresponding author upon reasonable request.

**Conflicts of Interest:** The authors declare no conflict of interest. The funders had no role in the design of the study; in the collection, analyses, or interpretation of data; in the writing of the manuscript; or in the decision to publish the results.

## Nomenclature

| Acronyms | Description |
| --- | --- |
| RES | Renewable energy sources |
| COP27 | 27th Conference of the Parties |
| IEA | International energy agency |
| BESS | Battery energy storage system |
| SoC | State of charge |
| VSC-HVDC | Voltage source converter—high voltage direct current |
| MMG | Multi-microgrid |
| MPC | Model predictive control |
| UT | Unscented transform |
| ADMM | Alternating direction method of multipliers |
| MVAC | Medium voltage alternating current |
| MPPT | Maximum power point tracking |
| PV | Photovoltaic |
| $\omega$ | Output frequency |
| $\omega_0$ | Nominal bus frequency |
| $m$ | Frequency drooping coefficient |
| $P$ | Active power |
| $P^*$ | Active power demand |
| $V$ | Voltage amplitude |
| $V_0$ | Nominal voltage |
| $n$ | Voltage drooping coefficient |
| $Q$ | Reactive power |
| $Q^*$ | Reactive power demand |
| $\Delta\omega$ | Frequency deviation |
| $\omega_{max}$ | Frequency maximum |
| $\omega_{high}$ | Frequency high |
| $\omega_{low}$ | Frequency low |
| $\omega_{min}$ | Frequency minimum |
| $\Delta\omega_{max}$ | Maximum frequency deviation |
| $\Delta\omega_{high}$ | High-frequency deviation |
| $SOC_{max}$ | Maximum state of charge |
| $SOC_{high}$ | High state of charge |
| $SOC_{low}$ | Low state of charge |
| $SOC_{min}$ | Minimum state of charge |
| $P_{pv}$ | PV power |
| $P_{pv}^*$ | PV power demand |
| $m_p^{pv}$ | Proportional droop control coefficient of PV |
| $m_i^{pv}$ | Integral droop control coefficient of the PV |
| $P_{MPPT}$ | Maximum power point tracking of PV power |
| $P_{aux}$ | Auxiliary power |
| $m_p^{aux}$ | Proportional droop control coefficient of the auxiliary unit |
| $m_i^{aux}$ | Integral droop control coefficient of the auxiliary unit |
| $m_p^{local}$ | Proportional droop control coefficient of local converter |
| $P_{local}$ | Output power of the local converter |
| $P_{local}^*$ | Local converter power demand |
| $k_p^{vdc}$ | Proportional droop control coefficient of the DC link |
| $k_i^{vdc}$ | Integral droop control coefficient of the DC link |
| $V_{dc}$ | DC link voltage |
| $V_{dc}^*$ | DC link voltage demand |
| $\omega_{global}$ | Output frequency of the global converter |
| $m_p^{global}$ | Proportional droop control coefficient of global converter |
| $P_{global}$ | Output power of the global converter |
| $P_{global}^*$ | Power demand of the global converter |

| | |
|---|---|
| $k$ | Slope of the power demand of the global converter |
| $\Delta\omega_{local}$ | Frequency deviation of the local converter |
| $P_L$ | Load at the global bus |
| $P_{\exp,i}$ | Power export of the *ith* microgrid |
| $P_{\exp,i}^*$ | Power export demand of the *ith* microgrid |
| $N$ | Total number of connecting global converters |
| $i$ | Number of microgrids |
| $P_{\exp,avg}^*$ | Average power export demand |

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
