# Peer review of "Power Flow Management of Interconnected AC Microgrids Using Back-to-Back Converters"

_electronics, doi:10.3390/electronics12183765_

Round 1

Reviewer 1 Report

In this manuscript, the authors present a high-level global droop controller, that exchanges an exact amount of power between interconnected microgrids. This work is interesting, but still needs some modification before the acceptance of Electronics:

1. Introduction is too long. This section should present the background and significance of this work to readers in a simple and quick way.

2. The order of figures in the manuscript is chaotic. The figures should be numbered sequentially.

3. The font size in Figure 3, 4, and 5 is too small to recognize.

4. Space is required between numbers and units, such as line 497.

5. Figure 6 displays the auxiliary power utilization for 30 days. Why does power show significant differences on different days? Three operating scenarios should be marked in Figure 6.

N/A

Author Response

Thank you for reviewing our paper manuscript.

Please, find attached the point-by-point response to the comments.

Reviewer 2 Report

The table with nomenclature is not alphabetical ordered. Some abbreviations are missing in the table e.g. "BESS". I think it should be corrected.

Author Response

Thank you for reviewing our paper manuscript.

Please, find attached our response to the comments.

Reviewer 3 Report

Dear authors,

the work needs to be corrected on a few points before being published.

 I have several notes on the table on the first page:

- It is not necessary to put COP27 and IEA in the table, but only the parts that are part of the model, the electrical components and the mathematics of the problem you set out to solve.

- Also add the acronym BESS

- Remove the acronym of MPC, because it is not the objective of the paper. Put it only in the text where it is referred to.

- Make two tables one for the acronyms and one for the parameters/variables of the equations.

- Put only "omega", do not put equations in the table

- It is not necessary to put the Laplace operator in the table.

- Put only 'i' without its respective incremental value.

The simulation needs to be more detailed.  In addition, I would like to ask some questions about the simulation.

- what kind of models were used for the simulink simulation for PV, RES, BESS?

- were the converters put in with the switches or were average models used?

Observations on the figures:

-they are poorly visible and quite unfocused. If you are using writing tools like Latex you can import them in eps format, they should read better.

-the data on the SoC/day are not very readable. Also, how did you simulate daily data?

References should be expanded on both intelligent microgrid management, e.g.: https://doi.org/10.1007/s40435-018-00506-5 and intelligent management of electrical systems in other contexts, e.g.: https://doi.org/10.3390/en11113216

Small changes:

-In the paper the style of the text or its spacing changes several times, fix this.

-Line 228: remove the semicolon after "where"

Thank you.

Author Response

Thank you for reviewing our paper manuscript.

Please, find attached our point-by-point response to the comments.

Round 2

Reviewer 3 Report

Dear Authors,

thank you for your answers.